# Neurosymbolic Reinforcement Learning with Formally Verified Exploration

**Greg Anderson**
UT Austin
`ganderso@cs.utexas.edu`

**Abhinav Verma**
UT Austin
`verma@utexas.edu`

**Isil Dillig**
UT Austin
`isil@cs.utexas.edu`

**Swarat Chaudhuri**
UT Austin
`swarat@cs.utexas.edu`

## Abstract

We present REVEL, a partially neural reinforcement learning (RL) framework for provably safe exploration in continuous state and action spaces. A key challenge for provably safe deep RL is that repeatedly verifying neural networks within a learning loop is computationally infeasible. We address this challenge using two policy classes: a general, neurosymbolic class with approximate gradients and a more restricted class of symbolic policies that allows efficient verification. Our learning algorithm is a mirror descent over policies: in each iteration, it safely lifts a symbolic policy into the neurosymbolic space, performs safe gradient updates to the resulting policy, and projects the updated policy into the safe symbolic subset, all without requiring explicit verification of neural networks. Our empirical results show that REVEL enforces safe exploration in many scenarios in which Constrained Policy Optimization does not, and that it can discover policies that outperform those learned through prior approaches to verified exploration.

## 1 Introduction

Guaranteeing that an agent behaves safely during exploration is a fundamental problem in reinforcement learning (RL) [13, 1]. Most approaches to the problem are based on stochastic definitions of safety [23, 7, 1, 7], requiring the agent to satisfy a safety constraint with high probability or in expectation. However, in applications such as autonomous robotics, unsafe agent actions — no matter how improbable — can lead to cascading failures with high human and financial costs. As a result, it can be important to ensure that the agent behaves safely *even on worst-case inputs*.

A number of recent efforts [3, 11] use formal methods to offer such worst-case guarantees during exploration. Broadly, these methods construct a space of provably safe policies *before* the learning process starts. Then, during exploration, a *safety monitor* observes the learner, forbidding all actions that cannot result from one of these safe policies. If the learner is about to take a forbidden action, a safe policy (a *safety shield*) is executed instead.

So far, these methods have only been used to discover policies over simple, finite action spaces. Using them in more complex settings — in particular, continuous action spaces — is much more challenging. A key problem is that these safety monitors are constructed a priori and are blind to the internal state of the learner. As we experimentally show later in this paper, such a "one-size-fits-all" strategy can unnecessarily limit exploration and impede learner performance.

In this paper, we improve this state of the art through an RL framework, called REVEL[1] that allows learning over continuous state and action spaces, supports (partially) neural policy representations and contemporary policy gradient methods for learning, while also ensuring that every intermediate policy that the learner constructs during exploration is safe on worst-case inputs. Like previous efforts, REVEL uses monitoring and shielding. However, unlike in prior work, the monitor and the shield are updated as learning progresses.

A key feature of our approach is that we repeatedly invoke a formal verifier from within the learning loop. Doing this is challenging because of the high computational cost of verifying neural networks. We overcome this challenge using a neurosymbolic policy representation in which the shield and the monitor are expressed in an easily-verifiable symbolic form, whereas the normal-mode policy is given by a neural network. Overall, this representation admits efficient gradient-based learning as well as efficient updates to both the shield and monitor.

To learn such neurosymbolic policies, we build on PROPEL [29], a recent RL framework in which policies are represented in compact symbolic forms (albeit without consideration of safety), and design a learning algorithm that performs a functional mirror descent in the space of neurosymbolic policies. The algorithm views the set of shields as being obtained by imposing a constraint on the general policy space. Starting with a safe but suboptimal shield, it alternates between: (i) safely *lifting* the current shield into the unconstrained policy space by adding a neural component; (ii) safely *updating* this neurosymbolic policy using approximate gradients; and (iii) using a form of imitation learning to *project* the updated policy back into the constrained space of shields. Importantly, none of these steps requires direct verification of neural networks.

Our empirical evaluation, on a suite of continuous control problems, shows that REVEL enforces safe exploration in many scenarios where established RL algorithms (including CPO [1], which is motivated by safe RL) do not, while discovering policies that outperform policies based on static shields. Also, building on results for PROPEL, we develop a theoretical analysis of REVEL.

In summary, the contributions of the paper are threefold. First, we introduce the first RL approach to use deep policy representations and policy gradient methods while guaranteeing formally verified exploration. Second, we propose a new solution to this problem that combines ideas from RL and formal methods, and we show that our method has convergence guarantees. Third, we present promising experimental results for our method in the continuous control setting.

## 2   Preliminaries

**Safe Exploration.**   We formulate our problem in terms of a Markov Decision Process (MDP) that has the standard probabilistic dynamics, as well as a worst-case dynamics that is used for verification. Formally, such an MDP is a structure $M = (\mathcal{S}, \mathcal{A}, P, c, \gamma, p_0, P^{\#}, \mathcal{S}_U)$. Here, $\mathcal{S}$ is a set of environment states; $\mathcal{A}$ is a set of agent actions; $P(s' \mid s, a)$ is a probabilistic transition function; $c : \mathcal{S} \times \mathcal{A} \to \mathbb{R}$ is a state-action cost function; $0 < \gamma < 1$ is a discount factor; $p_0(s)$ is an initial state distribution with support $\mathcal{S}_0$; $P^{\#} : \mathcal{S} \times \mathcal{A} \to 2^{\mathcal{S}}$ is a deterministic function that defines *worst-case bounds* on the environment behavior; and $\mathcal{S}_U$ is a designated set of *unsafe states* that the learner must always avoid. Because our focus is on continuous domains, we assume that $\mathcal{S}$ and $\mathcal{A}$ are real vector spaces. The function $P^{\#}$ is assumed to be available in closed form to the learner; because it captures worst-case dynamics, we require that $\text{supp}(P(s' \mid s, a)) \subseteq P^{\#}(s, a)$ for all $s, a$. In general, the method for obtaining $P^{\#}$ will depend on the problem under consideration. In Section 4 we explain how we generate these worst case bounds for our experimental environments.

A *policy* for $M$ is a (stochastic) map $\pi : \mathcal{S} \to \mathcal{A}$ that determines which action the agent should take in a given state. Each policy $\pi$ induces a probability distribution on the cost $c_i$ at each time step $i$. The aggregate cost of a policy $\pi$ is $J(\pi) = \mathrm{E}[\sum_i \gamma^i c_i]$, where $c_i$ is the cost at the $i$-th time step.

For a set $S \subseteq \mathcal{S}$, we define the set of states $\text{reach}_i(\pi, S)$ that are reachable from $S$ in $i$ steps under worst-case dynamics:

$$\text{reach}_1(\pi, S) = \bigcup\nolimits_{s \in S, a \in \text{supp}(\pi(\cdot \mid s))} P^{\#}(s, a) \qquad \text{reach}_{i+1}(\pi, S) = \text{reach}_1(\pi, \text{reach}_i(\pi, S)).$$

The policy $\pi$ is *safe* if $\left( \bigcup_i \text{reach}_i(\pi, \mathcal{S}_0) \right) \cap \mathcal{S}_U = \varnothing$. If $\pi$ is safe, we write $Safe(\pi)$.

**Algorithm 1** Reinforcement Learning with Formally Verified Exploration (REVEL)

---

1: **Input:** Symbolic Policy Class $\mathcal{G}$ & Neural Policy Class $\mathcal{F}$.
2: **Input:** Initial $g_0 \in \mathcal{G}$, with the guarantee $\phi_0 \vdash Safe(g_0)$ for some $\phi_0$
3: Define neurosymbolic policy class $\mathcal{H} = \{h(s) \equiv \textbf{if } P^\#(s, f(s)) \subseteq \phi \textbf{ then } f(s) \textbf{ else } g(s)\}$
4: **for** $t = 1, \ldots, T$ **do**
5:    $h_t \leftarrow \text{LIFT}_{\mathcal{H}}(g_t, \phi_t)$      *//lifting the new symbolic policy and proof into the blended space*
6:    $h_t \leftarrow \text{UPDATE}_{\mathcal{F}}(h_t, \eta)$      *//policy gradient in neural policy space with learning rate $\eta$*
7:    $(g_{t+1}, \phi_{t+1}) \leftarrow \text{PROJECT}_{\Pi}(h_t)$    *//synthesis of safe symbolic policy and corresponding invariant*
8: **end for**
9: **Return:** Policy $h_T$

---

We define a *learning process* as a sequence of policies $\mathcal{L} = \pi_0, \pi_1, \ldots, \pi_m$. We assume that the initial policy $\pi_0$ in this sequence is worst-case safe. Our algorithmic objective is to discover a learning process $\mathcal{L}$ such that the final policy $\pi_m$ is safe and optimal, and every intermediate policy is safe:

$$\pi_m = \underset{\pi \text{ s.t. } Safe(\pi)}{\arg\min} \ J(\pi) \tag{1}$$

$$\forall 0 \leq i \leq m : Safe(\pi_i). \tag{2}$$

**Formal Verification.** Our learning algorithm relies on an oracle for formal verification of policies. Given a policy $\pi$, such a verifier tries to construct an inductive proof of the property $Safe(\pi)$. Such a proof takes the form of an *inductive invariant*, defined as a set of states $\phi$ such that: (i) $\phi$ includes the initial states, i.e., $\mathcal{S}_0 \subseteq \phi$; (ii) $\phi$ is closed under the worst-case transition relation, i.e., $\text{reach}_1(\phi) \subseteq \phi$; and (iii) $\phi$ does not overlap with the unsafe states, i.e., $\phi \cap \mathcal{S}_U = \emptyset$. Intuitively, states $s$ in $\phi$ are such that even under worst-case dynamics, MDP trajectories from $s$ can never encounter an unsafe state. We use the notation $\phi \vdash \pi$ to indicate that policy $\pi$ can be proven safe using inductive invariant $\phi$.

Inductive invariants can be constructed in many ways. Our implementation uses *abstract interpretation* [9], which maintains some *abstract* state that approximates the *concrete* states which the system can reach. For example, the abstract state might be a hyperinterval in the state space of the program that defines independent bounds on each state variable. Critically, this abstract state is an *overapproximation*, meaning that while the abstract state may include states which are not actually reachable, it will always include *at least* every reachable state. By starting with an abstraction of the initial states and using abstract interpretation to propagate this abstract state through the environment transitions and the policy, we can obtain an abstract state which includes all of the reachable states of the system (that is, we compute approximations of $\text{reach}_i(\mathcal{S}_0)$ for increasing $i$). Then if this abstract state does not include any unsafe states, we can be sure that none of the unsafe states are reachable by any concrete trajectory of the system either.

## 3   Learning Algorithm

Our learning method is a functional mirror descent in policy space, based on approximate gradients, similar to PROPEL [29]. The algorithm relies on two policy classes $\mathcal{G}$ and $\mathcal{H}$, with $\mathcal{G} \subseteq \mathcal{H}$.

The class $\mathcal{G}$ comprises the policies that we use as shields. These policies are safe and can be efficiently certified as such. Because automatic verification works better on functions that belong to certain restricted classes and are represented in compact, symbolic forms, we assume some syntactic restrictions on our shields. The specific restrictions depend on the power of the verification oracle; we describe the choices made in our implementation in Section 3.1.

The larger class $\mathcal{H}$ consists of neurosymbolic policies. Let $\mathcal{F}$ be a predefined class of neural policies. We assume that each shield in $\mathcal{G}$ can also be expressed as a policy in $\mathcal{F}$, i.e., $\mathcal{G} \subseteq \mathcal{F}$. Policies $h \in \mathcal{H}$ are of the form:

$$h(s) = \textbf{if } (P^\#(s, f(s)) \subseteq \phi) \textbf{ then } f(s) \textbf{ else } g(s)$$

where $g \in \mathcal{G}$, $f \in \mathcal{F}$, and $\phi$ is an inductive invariant that establishes $Safe(g)$. We commonly denote a policy $h$ as above by the notation $(g, \phi, f)$.

The "true" branch in the definition of $h$ represents the normal mode of the policy. The condition $P^\#(s, f(s)) \subseteq \phi$ is the safety monitor. If this condition holds, then the action $f(s)$ is safe, as it can

---
**Algorithm 2** Implementation of $\text{PROJECT}_{\mathcal{G}}$

---
1: **Input:** A neurosymbolic policy $h = (g, \phi, f)$ where $g = [(g_1, \chi_1), \dots, (g_n, \chi_n)]$
2: $g^* \leftarrow g$

3: **for** $t = 1, \dots, T$ **do**
4:     $\psi \leftarrow \text{CUTTINGPLANE}(\chi_i)$ for heuristically selected $i$
5:     $g_i^1 \leftarrow \text{IMITATESAFELY}(f, g_i, \chi_i \wedge \psi)$;      $g_i^2 \leftarrow \text{IMITATESAFELY}(f, g_i, \chi_i \wedge \neg\psi)$
6:     $g' \leftarrow \text{SPLIT}(g, i, (g_i^1, \chi_i \wedge \psi), (g_i^2, \chi_i \wedge \neg\psi))$
7:     **if** $D(g', h) < D(g^*, h)$ **then**
8:        $g^* \leftarrow g'$
9:     **end if**
10: **end for**
11: $\phi^* \leftarrow \text{SAFESPACE}(g^*)$
12: **return** $(g^*, \phi^*)$

---

only lead to states in $\phi$ (which does not overlap with the unsafe states). If the condition does not hold, then $f$ can violate safety, and the shield $g$ is executed in its place. In either case, $h$ is safe. As for updates to $h$, we do not assume that the policy gradient $\nabla_{\mathcal{H}} J(h)$ in the space $\mathcal{H}$ exists, and approximate it by the gradient $\nabla_{\mathcal{F}} J(h)$ in the space $\mathcal{F}$ of neural policies.

We sketch our learning procedure in Algorithm 1. The algorithm starts with a (manually constructed) shield $g_0 \in \mathcal{G}$ and a corresponding invariant $\phi_0$, then iteratively performs the following steps.

**LIFT$_{\mathcal{H}}$.** This step takes as input a shield $g \in \mathcal{G}$ and its accompanying invariant $\phi$, and constructs the policy $(g, \phi, g) \in \mathcal{H}$. Note that the neural component of this policy is just the input shield $g$ (in a neural representation). In practice, to construct this component, we can train a randomly initialized neural network to imitate $g$, using an algorithm such as DAGGER [26]. Because the safety of any policy $(g, \phi, f')$ only depends on $g$ and $\phi$, this step is safe.

**UPDATE$_{\mathcal{F}}$.** This procedure performs a series of gradient updates to a neurosymbolic policy $h = (g, \phi, f)$. As mentioned earlier, this step uses the approximate policy gradient $\nabla_{\mathcal{F}} J(h)$. This means that after an update, the new policy is $(g, \phi, f - \eta \nabla_{\mathcal{F}} J(h))$, for a suitable learning rate $\eta$. As the update does not change $g$ and $\phi$, the new policy is provably safe. Also, we show later that, under certain assumptions, the regret introduced by our approximation of the gradient is bounded.

**PROJECT$_{\mathcal{G}}$.** This procedure implements the projection operation of mirror descent. Given a neurosymbolic policy $h = (g, \phi, f)$, the procedure computes a policy $g' \in \mathcal{G}$ that satisfies $g' = \arg\min_{g'' \in \mathcal{G}} D(g'', (g, \phi, f))$ for some Bregman divergence $D$. Along with $g'$, we compute an invariant $\phi'$ such that $\phi' \vdash Safe(g')$.

The computation of $g'$ can be naturally cast as an imitation learning task with respect to the demonstration oracle $(g, \phi, f)$. Prior work [29, 30] has given heuristic solutions to this problem for the case when $g''$ obeys certain syntactic constraints. In our setting, we have an additional semantic requirement: $g''$ must be provably safe. How to solve this problem depends on the precise definition of the class of shields $\mathcal{G}$. Section 3.1 sketches the approach to this problem used in our implementation.

### 3.1 Instantiation with Piecewise Linear Shields

Any attempt to implement REVEL must start by choosing a class $\mathcal{G}$ of shields. Policies in $\mathcal{G}$ should be sufficiently expressive to allow for good learning performance but also facilitate verification. In our implementation, we choose $\mathcal{G}$ to comprise *deterministic, piecewise linear policies* of the form:

$$g(s) = \begin{cases} g_1(s) & \text{if } \chi_1(s) \\ g_2(s) & \text{if } \chi_2(s) \wedge \neg\chi_1(s) \\ \dots \\ g_n(s) & \text{if } \chi_n(s) \wedge (\bigwedge_{1 \leq i < n} \neg\chi_i(s)), \end{cases}$$

where $\chi_1, \dots, \chi_n$ are linear predicates that *partition* the state space, and each $g_i$ is a linear function. We represent $g(s)$ as a list of pairs $(g_i, \chi_i)$. We refer to the subpart of the state space defined by $\chi_i \wedge \bigwedge_{j=1}^{i-1} \neg\chi_j$ as the *region* for linear policy $g_i$ and denote this region by $\text{Region}(g_i)$.

Now we sketch our implementation of Algorithm 1. Since the LIFT$_\mathcal{H}$ and UPDATE$_\mathcal{F}$ procedures are agnostic to the choice of $\mathcal{G}$, we focus on PROJECT$_\mathcal{G}$, which seeks to find a shield $g$ at minimum imitation distance $D(g, h)$ from a given $h \in \mathcal{H}$.

Our implementation of this operation is the iterative procedure in Algorithm 2. Here, we start with an input policy $h = (g, \phi, f)$. In each iteration, we identify a component $g_i$ with region $\chi_i$, then perform the following steps: (i) Sample a *cutting plane* that creates a more fine-grained partitioning of the safe region, by splitting the region $\chi_i$ into two new regions $\chi_i^1$ and $\chi_i^2$. (ii) For each new region $\chi_i^j$, use a subroutine IMITATESAFELY to construct a safe linear policy $g_i^j$ (and a corresponding invariant) that minimizes $D(g_i^j, h)$ within the region $\chi_i^j$. (iii) Replace $(g_i, \chi_i)$ by the two new components, leading to the creation of a new, refined shield $g'$. The procedure ends by returning the most optimal shield $g'$ (and an invariant obtained by combining the invariants of the $g_i^j$-s) constructed through this process.

Now we sketch IMITATESAFELY, which constructs safe and imitation-loss-minimizing linear policies. By collecting state-action pairs using DAGGER [26], we reduce the procedure's objective to a series of constrained supervised learning problems. Each of these problems is solved using a projected gradient descent (PGD) that alternates between gradient updates to a linear policy and projections into the set of safe linear policies. Critically, the constraint imposed on each of these optimization problems is constructed such that (i) the resulting policy is provably safe and (ii) the projection for the PGD algorithm is easy to compute. In our implementation these constraints take the form of a hyperinterval in the parameter space of the linear policies. We can then use abstract interpretation [9], a common framework for program verification, to prove that every controller within a particular hyperinterval behaves safely. For more details on IMITATESAFELY, see the supplementary material.

### 3.2 Theoretical Analysis

The REVEL approach introduces two new sources of error over standard mirror descent. First, we approximate the gradient $\nabla_\mathcal{H}$ by $\nabla_\mathcal{F}$, which introduces bias. Second, our projection step may be inexact. Prior work [29] has studied methods for implementing the projection step with bounded error. Here, we bound the bias in the gradient approximation under some simplifying assumptions, and use this result to prove a regret bound on the final shield that our method converges on. We define a safety indicator $Z$ which is zero whenever the shield is invoked and one otherwise. We assume:

1. $\mathcal{H}$ is a vector space equipped with an inner product $\langle \cdot, \cdot \rangle$ and induced norm $\|h\| = \sqrt{\langle h, h \rangle}$,
2. $J$ is convex in $\mathcal{H}$, and $\nabla J$ is $L_J$-Lipschitz continuous on $\mathcal{H}$,
3. $\mathcal{H}$ is bounded (i.e., $\sup\{\|h - h'\| \mid h, h' \in \mathcal{H}\} < \infty$),
4. $\mathbb{E}[1 - Z] \leq \zeta$, i.e., the probability that the shield is invoked is bounded above by $\zeta$,
5. the bias introduced in the sampling process is bounded by $\beta$, i.e., $\|\mathbb{E}[\widehat{\nabla}_\mathcal{F} \mid h] - \nabla_\mathcal{F} J(h)\| \leq \beta$, where $\widehat{\nabla}_\mathcal{F}$ is the estimated gradient
6. for $s \in \mathcal{S}$, $a \in \mathcal{A}$, and policy $h \in \mathcal{H}$, if $h(a \mid s) > 0$ then $h(a \mid s) > \delta$ for some fixed $\delta > 0$.

Intuitively, this last assumption amounts to cutting of the tails of the distribution so that no action can be arbitrarily unlikely. Now, let the variance of the gradient estimates be bounded by $\sigma^2$, and assume the projection error $\|g_t - g_t^*\| \leq \epsilon$ where $g_t^*$ is the exact projection of a neurosymbolic policy onto $\mathcal{G}$ and $g_t$ is the computed projection. Let $R$ be an $\alpha$-strongly convex and $L_R$-strongly smooth regularizer. Then the bias of our gradient estimate is bounded by Lemma 1 and the expected regret bound is given by Theorem 1.

**Lemma 1.** *Let $\gamma$ be the diameter of $\mathcal{H}$, i.e., $\gamma = \sup\{\|h - h'\| \mid h, h' \in \mathcal{H}\}$. Then the bias incurred by approximating $\nabla_\mathcal{H} J(h)$ with $\nabla_\mathcal{F} J(h)$ and sampling is bounded by*

$$\left\| \mathbb{E}\left[\widehat{\nabla}_t \mid h\right] - \nabla_\mathcal{H} J(h) \right\| = O(\beta + L_j \zeta).$$

**Theorem 1.** *Let $g_1, \ldots, g_T$ be a sequence of shields in $\mathcal{G}$ returned by REVEL and let $g^*$ be the optimal programmatic policy. Choosing a learning rate $\eta = \sqrt{\frac{1}{\sigma^2}\left(\frac{1}{T} + \epsilon\right)}$ we have the expected regret over $T$ iterations:*

$$\mathbb{E}\left[\frac{1}{T}\sum_{i=1}^{T} J(g_i)\right] - J(g^*) = O\left(\sigma\sqrt{\frac{1}{T} + \epsilon} + \beta + L_J\zeta\right)$$

This theorem matches the expectation that when a blended policy $h = (g, \phi, f)$ is allowed to take more actions without the shield intervening (i.e., $\zeta$ decreases), the regret bound is decreased. Intuitively, this is because when we use the shield, the action we take does not depend on the neural network $f$, so the learner does not learn anything useful. However if $h$ is using $f$ to choose actions, then we have unbiased gradient information as in standard RL.

## 4 Experiments

Now we present our empirical evaluation of REVEL. We investigate two questions: (1) How much safer are REVEL policies compared to state-of-the-art RL techniques that lack worst-case safety guarantees? What is the performance penalty for this increased safety? (2) Does REVEL offer significant performance gains over prior verified exploration approaches based on static shields[11, 3]?

To answer these questions, we compared REVEL against three baselines: (1) Deep Deterministic Policy Gradients (DDPG) [22]; (2) Constrained policy optimization (CPO) [1]; and (3) a variant of REVEL that never updates the user-provided shield. Of these, CPO is designed for safe exploration and takes into account a safety cost function. For DDPG, we engineered a reward function that has a penalty for safety violations. Details of hyperparameters that we used appear in the Appendix.

Our experiments used 10 benchmarks that include classic control problems, robotics applications, and benchmarks from prior work [11]. For each of these environments, we hand-constructed a worst-case, piecewise linear model of the dynamics. These models are based on the physics of the environment and use non-determinism to approximate nonlinear functions. For example, some of our benchmarks include trigonometric functions which cannot be represented linearly. In these cases, we define piecewise linear upper and lower bounds to the trigonometric functions. These linear approximations are necessary to make verification feasible. Each benchmark also includes a bounded-time safety property which should hold at all times during training.

**Performance.** First, we compare the policies learned using REVEL against policies learned using the baselines in terms of their cost (lower is better). Figures 1 and 2 show the cost over time of the policies during training. The results suggest that:

- The performance of REVEL is competitive with (or even better than) DDPG for 7 out of the 10 benchmarks. REVEL achieves significantly better reward than DDPG in the "car-racing" benchmark, and reward is only slightly worse for 2 benchmarks.

- REVEL has better performance than CPO on 4 out of the 10 benchmarks and only performs slightly worse on 2. Furthermore, the cost incurred by CPO is significantly worse on 2 benchmarks (noisy-road and car-racing).

- REVEL outperforms the static shielding approach on 4 out of 10 benchmarks. Furthermore, the difference is very substantial on two of these benchmarks (noisy-road and mountain-car).

REVEL does induce substantial overhead in terms of computational cost. The cost of the network updates and shield updates for each benchmark are shown in Table 1 along with the percentage of the total time spent in shield synthesis. The "acc" and "pendulum" benchmarks stand out as having very fast shield updates. For these two benchmarks the safety properties are relatively simple, so the verification engine is able to come up with safe shields more quickly. Otherwise, REVEL spends the majority of its time (87% on average) on shield synthesis.

**Safety.** To validate whether the safety guarantee provided by REVEL is useful, we consider how DDPG and CPO behave during training. Specifically, Table 2 shows the average number of safety violations per run for DDPG and CPO. As we can see from this table, DDPG and CPO both exhibit safety violations in 8 out of the 10 benchmarks. In Figure 3, we show how the number of violations varies throughout the training process for a few of the benchmarks. The remaining plots are left to the supplementary material.

**Qualitative Assessment.** To gain intuition about the difference between policies that REVEL and the baselines compute, we consider trajectories from the trained policies for two of our benchmarks that are easy to visualize. Figure 4 shows the trajectories taken by each of the policies for the obstacle2 benchmark. In this environment, the policy starts in the lower left corner, and the goal is to move to the green circle in the upper right. However, the red box in the middle is unsafe. As we can see from Figure 4, all of the policies have learned to go around the unsafe region in the center. However

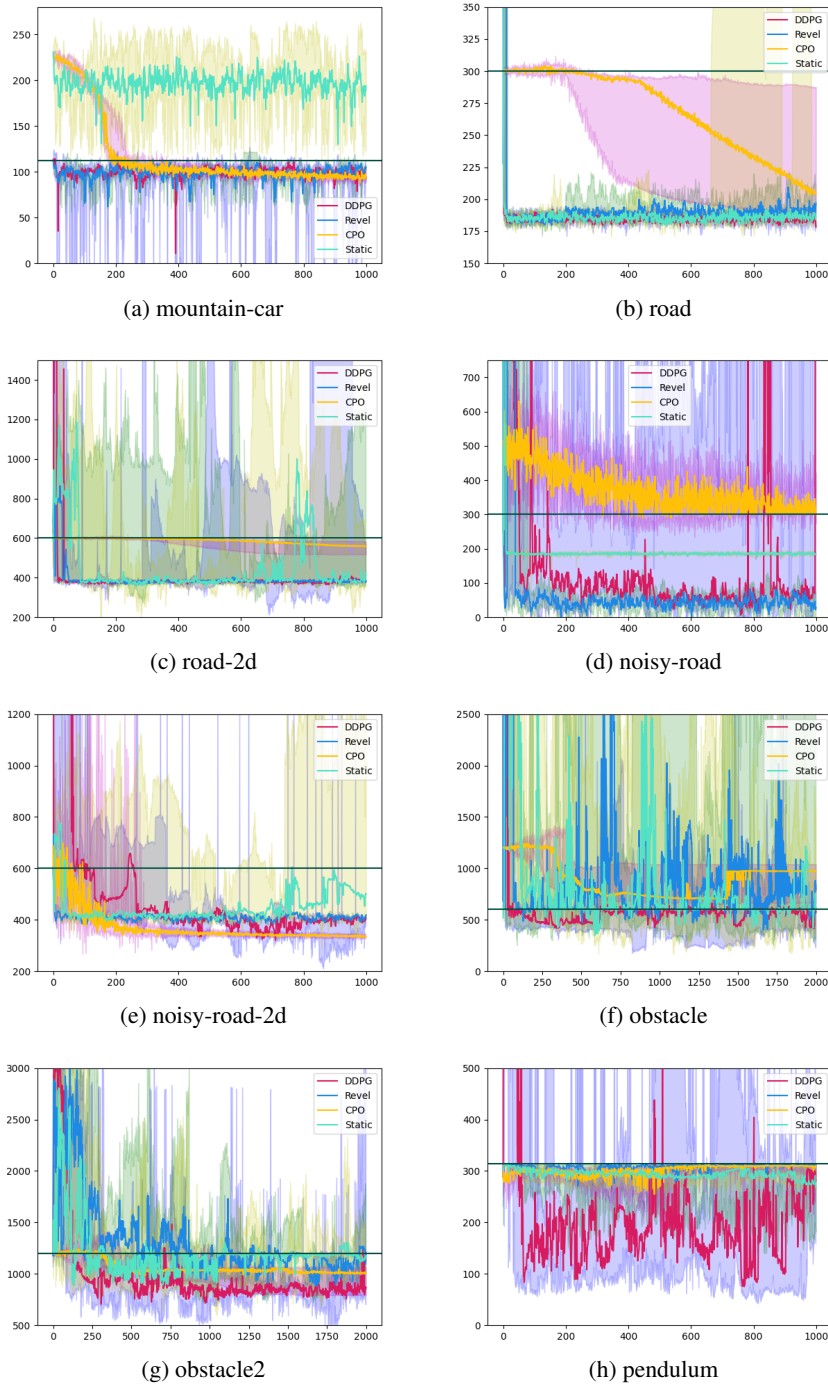

Figure 1: Training performance comparison on our benchmarks. The y-axis represents the Cost $J(\pi)$ and the x-axis gives the number of training episodes.

DDPG has not reinforced this behavior enough and still enters the unsafe region at the corner. By contrast, the statically shielded policy manages to avoid the region, but there is a very clear bend in its trajectory where the shield has to step in. Revel avoids the unsafe region while maintaining a smooth trajectory throughout. In this case, CPO also learns to avoid the unsafe region and go to the goal. (Because the environment is symmetrical, there is no significance to the CPO curve going up first and then right.)

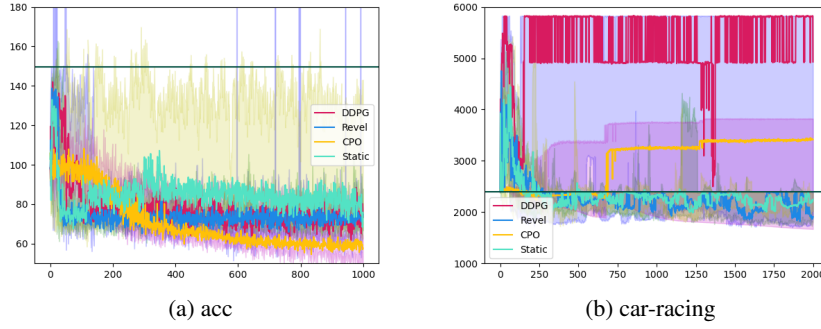

(a) acc                      (b) car-racing

Figure 2: Training performance comparison (continued). The y-axis represents the Cost $J(\pi)$ and the x-axis gives the number of training episodes.

Table 1: Training time in seconds for network and shield updates.

| Benchmark | Network update (s) | Shield update (s) | Shield percentage |
|---|---|---|---|
| mountain-car | 1900 | 5315 | 73.7% |
| road | 954 | 9401 | 90.8% |
| road-2d | 1015 | 19492 | 95.1% |
| noisy-road | 962 | 12793 | 93.0% |
| noisy-road-2d | 935 | 25514 | 96.5% |
| obstacle | 4332 | 27818 | 86.5% |
| obstacle2 | 4365 | 21661 | 83.2% |
| pendulum | 1292 | 113 | 8.0% |
| acc | 1097 | 56 | 4.9% |
| car-racing | 4361 | 15892 | 78.5% |

Table 2: Safety violations.

| Benchmark | DDPG | CPO |
|---|---|---|
| mountain-car | 0 | 3.6 |
| road | 0 | 0 |
| road-2d | 113.4 | 70.8 |
| noisy-road | 1130.4 | 8526.4 |
| noisy-road-2d | 107.4 | 0 |
| obstacle | 12.4 | 1.0 |
| obstacle2 | 96 | 118.6 |
| pendulum | 92.4 | 9906 |
| acc | 4 | 673 |
| car-racing | 4956.2 | 22.4 |

Figure 5 shows trajectories for "acc", which models an adaptive cruise control system where the goal is to follow a lead car as closely as possible without crashing into it. The lead car can apply an acceleration to itself at any time. The x-axis shows the distance to the lead car while the y-axis shows the relative velocities of the two cars. Here, all three trajectories start by accelerating to close the gap to the lead car before slowing down again. The statically shielded (and most conservative) policy is the first to slow down. The DDPG and CPO policies fail to slow down soon enough or quickly enough and crash into the lead car (the red region on the right side of the figure). In contrast, the REVEL policy can more quickly close the gap to the lead car and slow down later while still avoiding a crash.

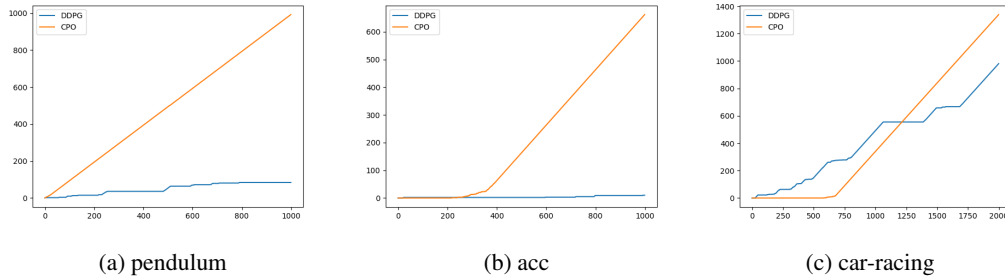

(a) pendulum            (b) acc            (c) car-racing

Figure 3: Cumulative safety violations during training.

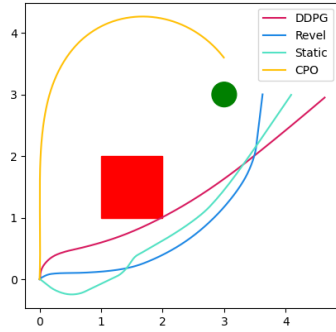

Figure 4: Trajectories for obstacle2.

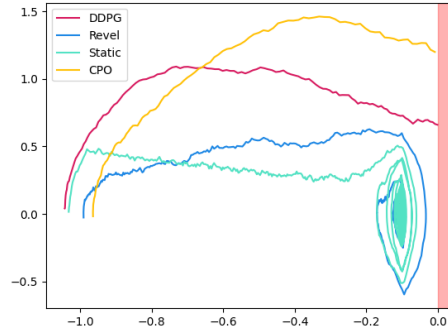

Figure 5: Trajectories for acc.

## 5 Related Work

There is a growing literature on safety in RL [14]. Approaches here can be classified on basis of whether safety is guaranteed during learning or deployment. REVEL, and, for example, CPO [1], were designed to enforce safety during training. Another way to categorize approaches is by whether their guarantees are probabilistic (or in expectation) or worst-case. Most approaches [23, 7, 1, 7] are in the former category; however, REVEL and prior approaches based on verified monitors [3, 11, 12] are in the latter camp. Now we discuss in more detail three especially related threads of work.

**Safety via Shielding.** These approaches rely on a definition of error states or fatal transitions to guarantee safety and have been used extensively in both RL and control theory [2, 3, 8, 11, 12, 16, 24, 32]. Our approach follows this general framework, but crucially introduces a mechanism to improve the shielded-policy during training. This is achieved by projecting the neural policy onto the shielded policy space. The idea of synthesizing a shield to imitate a neural policy has been explored in recent work [5, 32]. However, these approaches only generated the shield after training, so there are no guarantees about safety during training.

**Formal Verification of Neural Networks.** There is a growing literature on the verification of worst-case properties of neural networks [4, 15, 18, 19, 31]. In particular, a few recent papers [17, 27] target the verification of neural policies for autonomous agents. However, performing such verification inside a learning loop is computationally infeasible – in fact, state-of-the-art techniques failed to verify a single network from our benchmarks within half an hour.

**Safety via Optimization Constraints.** Many recent approaches to safe RL rely on specifying safety constraints as an additional cost function in the optimization objective [1, 6, 10, 20, 21]. These approaches typically provide safety up to a certain threshold by requiring that the additional cost function is kept below a certain constant. In contrast, our approach is suitable for use cases in which safety violations are completely unacceptable and where provable guarantees are required.

## 6 Conclusion

We have presented REVEL, an RL framework that permits formally verified exploration while supporting continuous action spaces and contemporary learning algorithms. Our key innovation is to cast the verified RL problem as a form of mirror descent that uses a verification-friendly symbolic policy representation along with a neurosymbolic policy representation that benefits learning.

One limitation of this work is its assumption of a fixed worst-case model of the environment. Allowing this model to be updated as learning progresses [12] is a direction for future work. The development of incremental verification techniques to further allay the cost of repeated verification is another natural direction. Progress on such verification techniques can potentially allow the use of more expressive classes of shields, which, in turn, can boost the learner's overall performance.

## Broader Impact

In the recent past, reinforcement learning has seen numerous advances and found applications in safety-critical settings. System failures in this setting can result in significant loss of property or even loss of life. This work takes a step towards solving this problem by guaranteeing that RL agents do not violate safety properties.

As with any safety-related work, the consequences of failure or misuse of this technique can be severe. Specifically, there is a risk that a user might assume that their system is guaranteed safe when this is not the case (for example, if the user fails to adequately specify the environment or safety property). Writing correct safety specifications is known to be hard, so inexpert users may feel an unwarranted sense of security. While misuse of the tool carries great risk, proper use can confer substantial advantages. In particular, it may allow the benefits of RL to be brought to domains, such as robotics and autonomous vehicles, where failure has a very high cost.

## Funding Acknowledgment

This work was supported in part by United States Air Force Contract # FA8750-19-C-0092, ONR award # N00014-20-1-2115, NSF Awards # CCF-2033851, # CCF-SHF-1712067, # CCF-SHF-1901376, and # CNS-CPS-1646522, and a JP Morgan Chase Fellowship (for Verma).

## Footnotes

[1] REVEL stands for **Re**inforcement learning with **ve**rified exp**l**oration. The current implementation is available at `https://github.com/gavlegoat/safe-learning`.

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
