[Supplementary Material]

# A  Safely Imitating a Neural Policy

In this section, we describe our projection algorithm for piecewise linear policies in more detail. Algorithm 2 defines this operation at a high-level, but leaves out some of the details of the IMITATE-SAFELY subroutine. The role of IMITATESAFELY is to learn a linear policy which is provably safe on some region and behaves as similarly to the neural controller as possible on that region. Since linear policies are differentiable, we adopt a projected gradient descent approach. To formalize this, we note that a linear policy $g$ is just a matrix in $\mathbb{R}^{n \times m}$ where $n$ is the dimension of the action space and $m$ is the dimension of the state space. We will use $\theta_g$ to refer to a vector in $\mathbb{R}^{nm}$ parameterizing $g$.

---

**Algorithm 3** Safely imitating a network using a given starting point and partition.

---

1: **Input:** A neural policy $f$, a region $\chi$, and a linear policy $g$.
2: $g^* \leftarrow g$
3: **while * do**
4:     $S \leftarrow \text{COMPUTESAFEREGION}(g^*, \chi)$
5:     $g^* \leftarrow g^* - \alpha \nabla D(g^*, N)$
6:     $g^* \leftarrow proj_S(g^*)$
7: **end while**
8: **return** $g^*$

---

Now our safe imitation algorithm is described in Algorithm 3. In each iteration, we first compute a safe region in the parameter space of $g^*$ over the region $\chi$. This is done by starting with a region bigger than the gradient step size and then iteratively searching for unsafe controllers and trimming the region to remove them. The returned region $S \subseteq \mathbb{R}^{n \times m}$ contains $g^*$ and only contains safe policies over the region $\chi$. This trimming process continues until $S$ can be verified using abstract interpretation [8]. In our implementation $S$ represents an interval in $\mathbb{R}^{nm}$ constraining $\theta_g$. Next, we take a gradient step according to the imitation loss $D$. For example $D$ may be computed using a DAgger-like algorithm to gather a dataset for supervised learning. Finally, we project $g^*$ into the safe region $S$ computed earlier. Specifically, this means projecting $\theta_{g^*}$ into the region of $\mathbb{R}^{nm}$ represented by $S$. Notice that since we project back into $S$ after each iteration, the policy returned by IMITATESAFELY is known to be safe on $\chi$.

Intuitively, recomputing $S$ at each iteration allows the controller to learn behavior which is more different from the starting point $g$ than would otherwise be possible. This is because computing a safe region involves abstracting the behavior of the system and in general it is intractable to compute the entire safe of safe policies in advance. Recomputing the safe space using the current controller at each step means we only need to prove the safety of a relatively small piece of the policy space local to the current controller. Specifically, if we can verify a region at least as large as one gradient step then the gradient descent procedure is unconstrained for that step. By repeating this process at each step, we only end up needing to verify a thin strip of policies surrounding the trajectory the gradient descent algorithm takes through the policy space.

# B  Theoretical Analysis

Here we provide proofs of the theoretical results from Section 3.2 and extend the discussion of a few theoretical issues.

Recall from Section 3.2 that we require the policy space $\mathcal{H}$ to be a vector space equipped with an inner product $\langle \cdot, \cdot \rangle$ inducing a norm $\|h\| = \sqrt{\langle h, h \rangle}$. Addition and scalar multiplication are defined in the standard way, i.e., $(\lambda u + \kappa v)(s) = \lambda u(s) + \kappa v(s)$. The cost functional of a policy $u$ is defined as $J(u) = \int_{\mathcal{S}} c(s, u(s)) d\mu^u(s)$ where $\mu^u$ is the state distribution induced by $u$. We assume that $\mathcal{G}$ and $\mathcal{F}$ are subspaces of $\mathcal{H}$ so that there is a well-defined notion of distance between policies in these classes. Additionally, notice that while a policies in $\mathcal{G}$ may not be differentiable in terms of their programmatic representation, they may still be differentiable when viewed as points in the ambient space $\mathcal{H}$. We will assume $\mathcal{H}$ is parameterizable by a vector in $\mathbb{R}^N$ for some $N$.

We will make use of a few standard notions from functional analysis, restated here for convenience:

**Definition 1.** *(Strong convexity) A differentiable function $R$ is $\alpha$-strongly convex w.r.t. a norm $\| \cdot \|$ if* $R(y) \geq R(x) + \langle \nabla R(x), y - x \rangle + \frac{\alpha}{2} \|y - x\|^2$.

**Definition 2.** *(Lipschitz continuous gradient smoothness) A differentiable function $R$ is $L_R$-strongly smooth w.r.t. a norm $\|\cdot\|$ if $\|\nabla R(x) - \nabla R(y)\|_* \leq L_R\|x - y\|$.*

**Definition 3.** *(Bregman divergence) For a strongly convex regularizer $R$, $D_R(x, y) = R(x) - R(y) - \langle \nabla R(y), x - y\rangle$ is the Bregman divergence between $x$ and $y$. Note that $D_R$ is not necessarily symmetric.*

With these preliminaries, we can now prove Theorem 1 from Section 3.2. The high-level strategy for this proof will be to prove Lemma 1, and then combine this result with a more general regret bound from [28]. First we restate the general theorem below. Let $R$ be an $\alpha$-strongly convex and $L_R$-smooth functional w.r.t. the norm $\|\cdot\|$ on $\mathcal{H}$. Additionally let $\nabla_{\mathcal{H}}$ be a Fréchet gradient on $\mathcal{H}$. Then our algorithm can be described as follows: start with $g_0 \in \mathcal{G}$ (provided by the user) then for each iteration $t$:

1. Compute a noisy estimate of the gradient $\widehat{\nabla} J(g_{t-1}) \approx \nabla J(g_{t-1})$.

2. Update in $\mathcal{H}$: $\nabla R(h_t) = \nabla R(h_{t-1}) - \eta \widehat{\nabla} J(g_{t-1})$.

3. Perform an approximate projection $g_t = \text{proj}_{\mathcal{G}}^R(h_t) \approx \arg\min_{g \in \mathcal{G}} D_r(g, h_t)$.

This procedure is approximate functional mirror descent under bandit feedback. We let $D$ be the diameter of $\mathcal{G}$, i.e., $D = \sup\{\|g - g'\| \mid g, g' \in \mathcal{G}\}$. $L_J$ is the Lipschitz constant of the functional $J$ on $\mathcal{H}$. $\beta$ and $\sigma^2$ are bounds on the bias and variance, respectively, of the gradient estimate in each iteration. $\alpha$ and $L_R$ are the strongly convex and smooth coefficients of the functional regularizer $R$. Finally $\epsilon$ is the bound on the projection error with respect to the same norm $\|\cdot\|$. We will make use of the following general result:

**Theorem 2.** *[28] Let $g_1, \ldots, g_T$ be a sequence of programmatic policies returned by REVEL and $g^*$ be the optimal programmatic policy. We have the expected regret bound*

$$\mathbb{E}\left[\frac{1}{T}\sum_{t=1}^{T} J(g_t)\right] - J(g^*) \leq \frac{L_R D^2}{\eta T} + \frac{\epsilon L_R D}{\eta} + \frac{\eta(\sigma^2 + L_J^2)}{\alpha} + \beta D.$$

*In particular, choosing $\eta = \sqrt{(1/T + \epsilon)/\sigma^2}$, this simplifies to*

$$\mathbb{E}\left[\frac{1}{T}\sum_{t=1}^{T} J(g_t)\right] - J(g^*) = O\left(\sigma\sqrt{\frac{1}{T} + \epsilon} + \beta\right).$$

Now we restate and prove Lemma 1 from the main paper to provide a bound on the bias of our gradient estimate. Recall our definition of the immediate safety indicator $Z$ as zero if the shield is invoked and one otherwise. Recall the assumptions from Section 3.2:

1. $J$ is convex in $\mathcal{H}$ and $\nabla J$ is $L_J$-Lipschitz continuous on $\mathcal{H}$,

2. $\mathcal{H}$ is bounded (i.e., $\sup\{\|h - h'\| \mid h, h' \in \mathcal{H}\} < \infty$),

3. $\mathbb{E}[1 - Z] \leq \zeta$, i.e., the probability that the shield is invoked is bounded above by $\zeta$,

4. the bias introduced in the sampling process is bounded by $\beta$, i.e., $\|\mathbb{E}[\widehat{\nabla}_{\mathcal{F}} \mid h] - \nabla_{\mathcal{F}} J(h)\| \leq \beta$, where $\widehat{\nabla}_{\mathcal{F}}$ is the estimated gradient

5. for $s \in \mathcal{S}$, $a \in \mathcal{A}$, and policy $h \in \mathcal{H}$, if $h(a \mid s) > 0$ then $h(a \mid s) > \delta$ for some fixed $\delta > 0$.

Under these assumptions:

**Lemma.** *Let $D$ be the diameter of $\mathcal{H}$, i.e., $D = \sup\{\|h - h'\| \mid h, h' \in \mathcal{H}\}$. Then the bias incurred by approximating $\nabla_{\mathcal{H}} J(h)$ with $\nabla_{\mathcal{F}} J(h)$ and sampling is bounded by*

$$\left\|\mathbb{E}\left[\widehat{\nabla}_{\mathcal{F}} \mid h\right] - \nabla_{\mathcal{H}} J(h)\right\| = O(\beta + L_J\zeta).$$

*Proof.* First, we note that $\|\mathbb{E}[\widehat{\nabla}_{\mathcal{F}} \mid h] - \nabla_{\mathcal{H}} J(h)\| \leq \|\mathbb{E}[\widehat{\nabla}_{\mathcal{F}} \mid h] - \nabla_{\mathcal{F}} J(h)\| + \|\nabla_{\mathcal{F}} J(h) - \nabla_{\mathcal{H}} J(h)\|$. We have already assumed that the first term is bounded by $\beta$, so we will proceed to bound the second term.

Let $h = (g, \phi, f)$ be a policy in $\mathcal{H}$. By the policy gradient theorem [27], we have that

$$\nabla_{\mathcal{F}} J(h) = \mathbb{E}_{s \sim \rho_h, a \sim h} \left[ \nabla_{\mathcal{F}} \log h(a \mid s) Q^h(s, a) \right] \tag{3}$$

where $\rho_h$ is the state distribution induced by $h$ and $Q^h$ is the long-term expected reward from a state $s$ and action $a$. We will omit the distribution subscript in the remainder of the proof for convenience. Now note that if $Z$ is one, then then $h(a \mid s) = f(a \mid s)$, so that in particular

$$\nabla_{\mathcal{F}} \log h(a \mid s) Q^h(s, a) = \nabla_{\mathcal{H}} \log h(a \mid s) Q^h(s, a).$$

On the other hand, if $Z$ is zero, then $h(a \mid s)$ is independent of $f$, and so we have

$$\nabla_{\mathcal{F}} \log h(a \mid s) Q^h(s, a) = 0.$$

Thus, we can rewrite Equation 3 as

$$\begin{aligned}
\nabla_{\mathcal{F}} J(h) &= \mathbb{E} \left[ Z \nabla_{\mathcal{H}} \log h(a \mid s) Q^h(s, a) \right] \\
&= \mathbb{E}[Z] \mathbb{E} \left[ \nabla_{\mathcal{H}} \log h(a \mid s) Q^h(s, a) \right] + \mathrm{Cov}(S, \nabla_{\mathcal{H}} \log h(a \mid s) Q^h(s, a)) \\
&= \mathbb{E}[Z] \nabla_{\mathcal{H}} J(h) + \mathrm{Cov}(S, \nabla_{\mathcal{H}} \log h(a \mid s) Q^h(s, a)). \tag{4}
\end{aligned}$$

Note that the covariance term is a vector where the $i$'th component is the covariance between $Z$ and the $i$'th component of the gradient $\nabla_{\mathcal{H}}^i$. Then for each $i$, by Cauchy-Schwarz we have

$$|\mathrm{Cov}(Z, \nabla_{\mathcal{H}}^i \log h(a \mid s) Q^h(s, a))| \leq \sqrt{\mathrm{Var}(Z) \mathrm{Var}(\nabla_{\mathcal{H}}^i \log h(a \mid s) Q^h(s, a))}.$$

Since $Z \in \{0, 1\}$ we must have $0 \leq \mathrm{Var}[Z] \leq 1$ so that

$$|\mathrm{Cov}(Z, \nabla_{\mathcal{H}}^i \log h(a \mid s) Q^h(s, a))| \leq \sqrt{\mathrm{Var}(\nabla_{\mathcal{H}}^i \log h(a \mid s) Q^h(s, a))}.$$

By assumption, for every state-action pair $(a, s)$ if $(a, s)$ is in the support of $\rho_h$ then $h(a \mid s) > \delta$. We also have that $Q^h(s, a)$ is bounded (because $J$ is Lipschitz on $\mathcal{H}$ and $\mathcal{H}$ is bounded). Then because the gradient of the log is bounded above by one and because $\nabla_H H$ is bounded by definition, we have $\|\nabla_{\mathcal{H}}^i \log h(a \mid s) Q^h(s, a)\|$ is bounded. Therefore by Popoviciu's inequality, $\mathrm{Var}(\nabla_{\mathcal{H}}^i \log h(a \mid s) Q^h(s, a))$ is bounded as well. Choose $B > \mathrm{Var}(\nabla_{\mathcal{H}}^i \log h(a \mid s) Q^h(s, a))$ for all $i$. Then we have $\|\mathrm{Var}(\nabla_{\mathcal{H}} \log h(a \mid s) Q^h(s, a))\|_\infty < \sqrt{B}$, and because $\mathcal{H}$ is finite-dimensional, $\|\mathrm{Var}(\nabla_{\mathcal{H}} \log h(a \mid s) Q^h(s, a))\| < c\sqrt{B}$ for some constant $c$ for any norm $\|\cdot\|$.

Substituting this into Equation 4, we have

$$\|\nabla_{\mathcal{F}} J(h) - \mathbb{E}[S] \nabla_{\mathcal{H}} J(h)\| < c\sqrt{B}.$$

Then

$$\begin{aligned}
\|\nabla_{\mathcal{F}} J(h) - \nabla_{\mathcal{H}} J(h)\| &\leq \|\nabla_{\mathcal{F}} J(h) - \mathbb{E}[S] \nabla_{\mathcal{H}} J(h)\| + \|\mathbb{E}[S] \nabla_{\mathcal{H}} J(h) - \nabla_{\mathcal{H}} J(h)\| \\
&< c\sqrt{B} + \|\mathbb{E}[S] \nabla_{\mathcal{H}} J(h) - \nabla_{\mathcal{H}} J(h)\|.
\end{aligned}$$

By assumption, $\nabla_{\mathcal{H}} J(h)$ is Lipschitz and $\mathcal{H}$ is bounded. Let $D$ be the diameter of $\mathcal{H}$ and recall that $L_J$ is the Lipschitz constant of $\nabla_{\mathcal{H}} J(h)$. Choose an arbitrary $h_0 \in \mathcal{H}$ and let $J_0 = \nabla_{\mathcal{H}} J(h_0)$. Then for any policy $h \in \mathcal{H}$ we have $\|\nabla_{\mathcal{H}} J(h)\| \leq J_0 + D L_J$. Then

$$\begin{aligned}
\|\mathbb{E}[Z] \nabla_{\mathcal{H}} J(h) - \nabla_{\mathcal{H}} J(h)\| &= \|(\mathbb{E}[Z] - 1) \nabla_{\mathcal{H}} J(h)\| \\
&= |\mathbb{E}[Z] - 1| \|\nabla_{\mathcal{H}} J(h)\| \\
&\leq |\mathbb{E}[Z] - 1| (J_0 + DL).
\end{aligned}$$

Since $Z$ is an indicator variable, we have $0 \leq \mathbb{E}[Z] \leq 1$ so that $|\mathbb{E}[Z] - 1| = 1 - \mathbb{E}[Z]$. Then finally we assume $D$ is a known constant to simplify presentation, and arrive at

$$\|\nabla_{\mathcal{F}} J(h) - \nabla_{\mathcal{H}} J(h)\| < c\sqrt{B} + (1 - \mathbb{E}[Z])(J_0 + DL_J) = O(L_J \zeta)$$

and plugging this back into the original triangle inequality we have

$$\left\| \mathbb{E} \left[ \widehat{\nabla}_{\mathcal{F}} \mid h \right] - \nabla_{\mathcal{H}} J(h) \right\| = O(\beta + L_J \zeta).$$

$\square$

Now Theorem 1 follows directly by plugging this bound on gradient estimate bias into Theorem 2.

## C   Experimental Data and Additional Results

In this section we provide more details about our experiments along with additional results.

First, we give a qualitative description of each benchmark:

- mountain-car is a continuous version of the classic mountain car problem. In this environment the goal is to move an underpowered vehicle up a hill by rocking back and forth in a valley to build up momentum. The safety property asserts that the car does not go over the crest of the hill on the left.

- road, road-2d, noisy-road, and noisy-road-2d are all variants of an autonomous car control problem. In each case, the car's goal is to move to a specified end position while obeying a given speed limit. The noisy variants introduce noise in the environment, while the 2d variants involve moving in two dimensions to reach the goal.

- In obstacle and obstacle2, a robot moving in 2D space must reach a goal position while avoiding an obstacle. In obstacle this obstruction is placed off to the side so it only affects the agent during exploration (but the shortest path to the goal does not intersect it). In obstacle2, the obstruction is placed between the starting position and the goal so that the policy must learn to move around it (see Figure 4).

- pendulum is a classic pendulum environment where the system must swing a pendulum up until it is vertical. The safety property in this case is a bound on the angular velocity of the pendulum.

- acc is an adaptive cruise control benchmark taken from [10] and modified to use a continuous action space. Here the goal is to follow a lead car as closely as possible without crashing into it. At each time step the lead car chooses an acceleration at random (from a truncated normal distribution) to apply to itself.

- car-racing is similar to obstacle2 except that in this case the goal is to reach a goal state on the opposite side of the obstacle and then come back. This requires the agent to complete a loop around the obstacle.

For each benchmark, we consider a bounded-time variant of the desired safety property. That is, for some fixed $T$ we guarantee that a policy $h = (g, \phi, f)$ cannot violate the safety property within $T$ time steps starting from any state satisfying $\phi$.

For most benchmarks, we train for 100,000 environment interactions with a maximum episode length of 100. For mountain-car we use a maximum episode length of 200 and 200,000 total environment interactions. For obstacle, obstacle2, and car-racing we use an episode length of 200 with 400,000 total environment interactions. For every benchmark we synthesize five new shields at even intervals throughout training. To evaluate CPO we use the implementation provided with the Safety Gym repository [24]. To account for our safety critical benchmarks, we reduce the tolerance for safety violations in this implementation by lowering the corresponding hyper-parameter from 25 to 1. For DDPG, we use an implementation from prior work [31], which is also what we base the code for REVEL on. We ran each experiment with five independent, randomly chosen seeds. Note that the chosen number of training episodes was enough for the baselines to converge, in the sense that over the last 25 training episodes we see less than a 2% improvement in policy performance.

We now provide more details about the safety violations seen during training. The plots in Figure 6 show the number of safety violations over time for DDPG and CPO. This figure is the same as Figure 3 except that it shows information for every benchmark.

(a) mountain-car

(b) road

(c) road-2d

(d) noisy-road

(e) noisy-road-2d

(f) obstacle

(g) obstacle2

(h) pendulum

(i) acc

(j) car-racing

Figure 6: Safety violations over time.