[Reviews · NeurIPS 2020]

Review 1

Summary and Contributions: This paper introduces a new framework for safe RL via shielding for continuous state and action spaces that trains an agent to develop formally verified policies that fulfil safety constrains both during training and test. The authors design a new methodology whose novelty relies on updating during training the shield that prevents the agent from selecting unsafe actions. The authors claim that this update mechanism helps their framework to outperform previous safe-aware approaches.

Strengths: The paper generally provides a detailed theoretical grounding of its claims. It's a novel approach, and to the best of my knowledge, it would be the first work introducing a computationally affordable framework that provides safety warranties during the whole training process while achieving a good performance in well-known complex benchmarks

Weaknesses: The proposed approach relies on a fixed worst-case model of the environment. It requires of an oracle for formal verification of policies and requires a manually constructed shield to start the learning process. Authors claim that their approach is computationally good, however, no empirical evidence is proved

Correctness: The technical claims of this paper seems to be generally correct. Still it misses a comparison which is significant for the evaluation of this contribution. I believe this work should include the computing time required by the proposed approach compared with the three baselines, specially since the authors claim several times the feasibility of their approach compared to previous ones based on Formal Methods. This will help to understand the trade-off between having the shield vs not having it and how much computing impact has updating the shield. I could raise my score for this submission if this comparison is added in the final version.

Clarity: The paper is well written. However, some parts required several reads to understand and I missed some running examples to illustrate the whole learning process. I think it would help potential readers to quickly grasp the key concepts of this paper.

Relation to Prior Work: Prior work seems to be correctly contrasted and referred, which helped to understand the contributions of this work. Well done!

Reproducibility: Yes

Additional Feedback: I think this is a good paper that should be accepted, the authors did a good submission. It just misses the additions I mentioned above to support the author claims about computational feasibility. Some minor comments: * In line 17, reference 6 is cited twice * Graphs in figures 1 and 2 have a horizontal red line that is not referred in the legend ----- Post-Rebuttal comments I appreciate the author efforts to answer my concerns. I still suggest authors to include not only the total computing time of their approach, but rather a comparison between the computing time required by a static shield against the proposed approach.


Review 2

Summary and Contributions: This paper consider a safe reinforcement learning problem by leveraging the recently developed neurosymbolic method. Particularly, in the exploration, a monitor P\sharp(s,f(s) ) \subset \phi observes the agent. If the policy f(s) is safe, then the agent just follows it .Otherwise it applies the policy in the shield. In the learning step, the agent can update the policy using any policy gradient method. Then it projects its updated policy on the space of the symbolic policy and get corresponding invariant \phi (some safe state) using algorithm 2. The author tests this algorithm in the several environment and shows its superiority over DDPG and CPO in the criteria of average cost and safety violation.

Strengths: It is an interesting work to combine the neural symbolic reinforcement learning with formal verification to solve the safe RL problem. I am not familiar with formal verification but the empirical result looks good. #######After response Thanks for the author's explanation, which resolves most of my concerns.

Weaknesses: 1. Some part of this paper is not clearly explained. For example, I guess people (at least for me ) in RL community are not familiar with formal verification. Can you give some simple example to intuitively explain how and why it works? 2. In section 3.1. the shield has the form of the piecewise linear policy. Does it work for complicated scenarios where the dimension of the state space is high? In that case the number of region is quite large. 3. In section 2, one assumption is that P\sharp is available in a closed form to the learner. Can you explain why it is reasonable? or give an example of P\sharp in your experiment.

Correctness: The algorithm and experiment result look correct and reasonable.

Clarity: The section of formal verification need to be further explained. See my comments on the weakness of this paper.

Relation to Prior Work: It clearly states the contribution of this paper and discuss the difference from previous literatures.

Reproducibility: Yes

Additional Feedback:


Review 3

Summary and Contributions: This paper proposes REVEL, a RL framework with verifiable safety guarantees. Neural networks are hard to verify, while symbolic policies are hard to directly optimize. To address this dilemma, it first constructs safe symbolic policies as shields. To facilitate learning, it lifts those policies into the neural-symbolic space and doing updates on them. In order to verify the new updated policies, it projects the policies back into the symbolic space. Theoretical results show that on expectation, by doing so, we will achieve lower and lower regrets. Empirical results claim that their method achieves competitive results against existing alternatives.

Strengths: (+) One of the reasons the prevents RL from being largely adopted in many real-world critical tasks is that it could lead to catastrophic failures. Considering this, this paper could be highly appealing to people that have ambitions in those domains. (+) The theoretical part is interesting. The results are quite clear and clean. Furthermore, they fit our intuition and well corroborate this paper’s claim. Yet, some analysis’s details can be more polished, which I will elaborate on in the next weakness box. (+) The paper is generally well organized. The supplementary also provide interesting details.

Weaknesses: (-) I have two majors concerns, one regarding the (theoretical) analysis and the other empirical evaluations. Speaking on the first point, it seems like all the safety guarantees boil down to the fact that the initial shields are safe and verifiable. However, when it gets transformed into the neural space, we use imitation learning and based my understanding, there is no guarantee that by imitation learning, the neural network would exactly reproduce what would happen in the shield. Granted, the initial symbolic-form shield is safe. Yet this transformation step seems to raise the possibility of unsafety. (-) On the empirical side. Based on my understanding, one major advantage of the proposed method is that the shield part can grow, which means the set of known safe policies are not a static concept. However, the current empirical evaluations seem not enough to justify this point. I believe a dedicated evaluation to measure what is the percentage that the f policy is executed could be an interesting thing to know. (-) It seems on Mountain-Car, the static version completely outperforms the proposed dynamic method, which is very much against the intuition and the theorem in this paper. I suppose the static and the proposed method receive the same initial shield. Then if the proposed method can keep growing the space of shield, then at least the proposed method should perform comparably to the static version. I understanding asking a method to achieve the best performance on every task is ridiculous. Yet, I believe proving analysis on failure case could shed very important light into better understanding what is the limitation of the method and why this method really works in practice. (-) This point is related to the first comment. Given my concern about the unsafety of imitation learning, I believe it could also be useful to show the number of safety violations of the proposed method. I suppose it should be all zeros. (-) This is a minor point. Assuming P# is available seems very unrealistic on real-world problems. I understand this is a necessary assumption to do analysis and give safety guarantees. Yet, what is a practical workaround considering when we are tackling a completely unknown domain.

Correctness: The theoreical analysis should be correct.

Clarity: This paper is in general well written and easy to follow.

Relation to Prior Work: The related work is well discussed.

Reproducibility: Yes

Additional Feedback: **** after rebuttal *** After reading the rebuttal, and more importantly, discussing with other reviewers, my concerns regarding the safety are fully resolved. I agree with other reviewers that an experiment about the computation time would be interesting.


Review 4

Summary and Contributions: The paper proposes a new algorithm for safe exploration in reinforcement learning. The paper proposes a policy representation that is easily verifiable for safety, and a way to update such policies. The paper studies the convergence of the algorithm under some special conditions, and shows promising experimental results in a range of continuous control problems.

Strengths: (i) I think the algorithm involves some interesting ideas, such as the policy representation and update method, that may be interesting to the safe RL community. - The experimental results look promising.

Weaknesses: (i) Parts of the algorithm remain unclear to me: - It is unclear to me how to specify inputs to the algorithm. How to initialize g? How to initially come up with a shield that is safe? How to specify P#, for example, in your computational experiments? - How is \phi computed? How expensive is it to compute \phi? (ii) There is no discussion about the algorithm's computational complexity. (iii) The piecewise linear policy class seems like a good example, but feels too limited as the only example. In the introduction, the authors claim that their approach uses deep policy representations. It would be great to have an example of a neural network policy class.

Correctness: I have not checked the appendix carefully, but the claims seem okay.

Clarity: I think the paper is pretty well written in general. A few minor suggestions: - Please label the axes of your figures. - Line 85, (ii), \phi is closed with respect to what policy? - Line 166, what policy is the expectation with respect to?

Relation to Prior Work: Yes.

Reproducibility: Yes

Additional Feedback: The setting considered seems equivalent to minimizing expected cost if the cost of being in an unsafe state is set to high enough. I wonder whether there is a specific benefit of the proposed algorithm compared to algorithms that are designed to minimize expected cost. In particular, I wonder if algorithms like MuZero, which plans many steps ahead, would be a good option here. *** The rebuttal addresses my concerns.

[Author Response · NeurIPS 2020]

First, we would like to thank the reviewers for their helpful feedback. Here, we address the main questions, and we will clarify all of these points in any future revision.

**Reviewers 1, 2, 3, and 4:**

- **It is difficult/impossible to generate a worst case environment model in many scenarios.** A formal proof of an agent's safety is a worst-case guarantee on states that will be reached at future time points. Such states are not well-defined without a worst-case environment model. For this reason, all prior work on formal approaches to safety in reinforcement learning (Alshiekh et al., 2018; Fulton & Platzer, 2018; Zhu et al., 2019) have required such models. As for how easy it is to write such models, note that the model only needs to capture worst-case constraints on the MDP, as opposed to the entirety of the agent's dynamics. For example, our adaptive cruise control benchmark uses a model of cars on a road and only constrains the maximum braking and accelerating force which can be generated by the cars. Writing such worst-case constraints by hand is often feasible. It may be possible to learn likely worst-case models through exploration, but this is a challenging orthogonal problem, and we leave it for future work.

- **How efficient is our algorithm?** The training time for our technique is typically about 8-10 hours using a 6-core desktop machine (and a single thread for shield updates). To clarify, our claim about efficiency is compared to approaches that provide guaranteed safety during training. Verifying purely neural representations in this context is prohibitively expensive. In order to validate this claim, we attempted to verify a network in one of our simpler benchmarks using Marabou (Katz et al., 2019), a state-of-the-art neural network analysis tool. We were unable to verify a property within half an hour. Since we need to make thousands of calls to the verification oracle during training, this indicates that a purely neural approach is infeasible for safe reinforcement learning. We will be happy to add more details on this in the paper's final version.

**Reviewer 2:**

- **Formal methods aspects of the paper are terse and hard to understand for the intended audience**. We will expand the relevant parts of the paper to give better intuition for the formal methods components of our algorithm.

- **Can shield synthesis scale to more complex state spaces?** Scalability depends on the number of regions in the piecewise linear policies (which the user can control) and underlying tools for formally verifying piecewise linear functions. These tools are advancing rapidly, and our framework can make use of new developments in this area.

**Reviewer 3:**

- **How is safety maintained by using imitation learning on the neural component of the policy?** Our policies have the form "if $\phi$ then $f$ else $g$" ($f$ is the neural component, $g$ is the shield), and the safety of the policy only depends on $\phi$ and $g$. The imitation learning part affects $f$, but not $g$ or $\phi$, and doesn't compromise the property above. Thus, the neurosymbolic policy is still safe, and therefore there are no safety violations in the experiments.

- **How often is the shield invoked?** Typically once training has finished the shield is not invoked. For example the "pendulum", "acc", and "mid-obstacle" benchmarks all have zero shield interventions after training. Intuitively this is because the network learns that running into the shield yields a lower reward. Earlier in training while the network is taking riskier exploration steps the shield is invoked more often, sometimes in as much as 20% of the time.

- **Why does the static shielding approach seem to outperform Revel on the mountain-car benchmark?** There appears to be a misunderstanding. In our plots, we show a loss rather than a reward. Thus lower is better on our plots, and Revel is significantly better than the static shielding approach.

**Reviewer 4:**

- **How do we initialize the shield and monitor?** In our experiments, the initial shields are trivial shields that one can easily write by hand. For example, in the obstacle benchmark, the shield never moves the controlled robot. In principle, we could adapt approaches from prior work (Fulton & Platzer, 2018; Zhu et al. 2019) to generate an initial shield automatically. However, this is itself a challenging, orthogonal question that is best left to future work.

- **How do we compute the monitor?** Our shield update algorithm produces a shield which is safe for at least the same states as the original shield. We then iteratively expand the monitor until we can no longer verify safety.

- **Theoretical complexity of the shield synthesis algorithm is not presented.** The shield synthesis algorithm used in the experiments is a form of imitation learning and runs for a chosen number of iterations before being cut off.

- **How is this approach different from minimizing the expected reward where unsafe states are given a large cost?** We prove that the system will *never* take any unsafe action (as opposed to giving a guarantee in expectation). In contrast, prior techniques that use the reward function to encode safety constraints must always explore a state at least once to measure the reward of that state. Specifically, MuZero relies on a learned prediction function which would itself need to be verified against the model in order to get the kind of guarantees we provide.

- **You only evaluate on piecewise linear policies but claim that the approach handles deep neural policies.** The policies in our experiments have the form "if $\phi$ then $f$ else $g$" where $g$ is piecewise-linear and $f$ is a neural network. We could replace $g$ with a different class of policies, but $g$ could not be a neural network because existing approaches for neural network for verification are not efficient enough to be incorporated into our safe exploration method.

[Meta-Review · NeurIPS 2020]

This paper introduces an RL method that satisfies safety constraints during both training and evaluation, via shielding for continuous state and action spaces so that unsafe actions are not selected. The main technical contribution is that there is a symbolic safety specification and policy, which is lifted in a continuous space via imitation learning. Policy updates occur in the lifted space, and then the policy is projected back to a symbolic space where verification can occur. The method has the added advantage that the definition of symbolic safe policies and safety specifications can increase over time as more experience is collected from interactions with the environment. I think this is an interesting scheme, and there are not many safe RL methods that can guarantee safety during training while expanding the safe set. A major drawback of the method is that it seems very time-consuming to update the policy, which was the main concern of the reviewers in the post-rebuttal discussion, but we all think that's a separate issue that doesn't detract from the merits of the method. Reviewers recommend a borderline accept. I agree and would recommend a poster.